# The Nutrient–Skin Connection: Diagnosing Eating Disorders Through Dermatologic Signs

**DOI:** 10.3390/nu16244354

**Published:** 2024-12-17

**Authors:** Efstathios Rallis, Kleomenis Lotsaris, Vasiliki-Sofia Grech, Niki Tertipi, Eleni Sfyri, Vassiliki Kefala

**Affiliations:** 1Department of Biomedical Sciences, School of Health and Care Sciences, University of West Attica, GR-12243 Athens, Greece; erallis@uniwa.gr (E.R.); ntertipi@uniwa.gr (N.T.); elsfiri@uniwa.gr (E.S.); valiakef@uniwa.gr (V.K.); 2Psychiatrist in Department of Psychiatry, Athens General Hospital ‘Evaggelismos’, GR-10676 Athens, Greece; psych.kleolots@gmail.com

**Keywords:** eating disorders, anorexia nervosa, bulimia nervosa, minerals, vitamins, starvation, self-induced vomiting, Russell’s sign

## Abstract

The interplay between nutrition and skin health provides a crucial lens for understanding, diagnosing, and managing eating disorders (EDs) such as anorexia nervosa (AN), bulimia nervosa (BN), and binge-eating disorder (BED). This review explores the dermatological manifestations resulting from the nutritional deficiencies commonly associated with EDs, including conditions like hair loss, xerosis, and brittle nails. These changes in the skin and its appendages often reflect deeper systemic dysfunctions, such as deficiencies in essential micronutrients (zinc, iron, and vitamins A and C), hormonal imbalances, and electrolyte disturbances. Recognizing these dermatological signs as diagnostic tools is vital for the early identification and intervention of EDs. By integrating dermatological observations with psychiatric and nutritional care, a holistic, multidisciplinary approach can be developed to address both the physical and psychological complexities of EDs. This review highlights the critical role of these skin-related markers in promoting timely diagnosis and effective treatment. To examine the relationship between specific nutrients and dermatological manifestations in EDs, a systematic review of three electronic databases—PubMed, Google Scholar, and ResearchGate—was conducted. The findings underline the importance of early recognition of these skin symptoms for effective management. Collaborative care involving dermatologists, psychiatrists, and nutritionists is essential for diagnosing and treating EDs. Such integrated efforts ensure a comprehensive approach to these multifaceted conditions, ultimately improving patient outcomes and enhancing overall care.

## 1. Introduction

The skin, the body’s largest organ, serves as a visible reflection of internal health, with changes in its appearance and function often indicating nutritional imbalances. In the setting of EDs—a spectrum of psychiatric conditions including AN, BN, and BED—nutritional deficiencies are both a cause and a consequence of disordered eating behaviors [1]. These deficiencies, ranging from micronutrient depletion to macronutrient imbalances, visibly manifest in the skin, hair, and nails, offering critical diagnostic insights.

Malnutrition, whether stemming from restrictive eating, purging, or other maladaptive behaviors, disrupts the body’s metabolic and structural integrity. Essential nutrients, such as zinc, iron, vitamins A and D, and essential fatty acids, are particularly susceptible to depletion [2]. This leads to distinct dermatological manifestations: zinc deficiency can cause psoriasis-like dermatitis and alopecia, iron deficiency results in brittle nails and telogen effluvium (hair thinning), hypovitaminosis A presents as xerosis (dry skin) and hyperkeratosis, while vitamin C deficiency impairs collagen synthesis, resulting in delayed wound healing and petechiae [3,4].

These nutrient-driven dermatological changes are not merely superficial but also reflect systemic dysfunctions. They often coincide with hormonal imbalances, electrolyte disturbances, and immune suppression, exacerbating the physical burden of EDs. Recognizing these signs provides clinicians with valuable tools for early detection, particularly in patients who might deny or downplay their condition. Subtle dermatologic markers, when accurately interpreted, can serve as indicators of underlying eating disorders, facilitating timely intervention [5,6].

This review delves into the intricate relationship between nutrient deficiencies and dermatologic manifestations in EDs. By examining how specific nutrient deficits contribute to visible skin changes, it underscores the diagnostic value of these markers. Furthermore, it highlights the importance of a multidisciplinary approach, integrating dermatological, psychiatric, and nutritional perspectives to address the complex interplay of physical and psychological factors in eating disorders [7]. This comprehensive strategy is essential for improving patient outcomes and ensuring holistic care.

## 2. Classification of Eating Disorders

The classification of eating disorders, as outlined in the Diagnostic and Statistical Manual of Mental Disorders, Fifth Edition, Text Revision (DSM-5-TR), identifies eight categories of feeding and eating disorders: AN, BN, BED, pica, rumination disorder (RD), avoidant/restrictive food intake disorder (ARFID), other specified feeding or eating disorder (OSFED), and unspecified feeding or eating disorder (UFED). Each disorder is defined by specific diagnostic criteria detailed in the DSM-5-TR. These include patterns of abnormal eating behaviors, distress or impairment in psychosocial functioning, and, for some disorders, physical health consequences. Disorders such as AN and BN involve disturbances in body weight perception and compensatory behaviors, while BED emphasizes recurrent binge eating without compensatory behaviors. Disorders like pica and RD focus on atypical consumption or regurgitation behaviors, respectively. ARFID is characterized by restrictive eating not linked to body image concerns. OSFED and UFED offer diagnostic adaptability for situations where symptoms do not fully align with the criteria for a specific eating disorder yet result in substantial distress or functional impairment. This comprehensive framework ensures tailored recognition and treatment for various manifestations of disordered eating [1].

The diagnostic criteria for these disorders according to DSM-5-TR [1] are outlined in Table 1 below.

## 3. Prevalence of Eating Disorders

EDs are prevalent across diverse populations, with notable variations in their occurrence and characteristics. Pica, for instance, is estimated to impact around 5% of school-aged children and approximately one-third of pregnant women, particularly those facing food insecurity and limited access to affordable, nutritious food. Although data on RD are sparse, European studies estimate its prevalence at 1–2% among grade-school-aged children, with higher rates observed in individuals with intellectual disabilities. Similarly, the prevalence of ARFID remains under-researched; however, one Australian study described a prevalence of 0.3% among people aged 15 and older [1].

A 2019 systematic review provided a broader understanding of feeding and eating disorders, estimating a lifetime prevalence of 8.4% in women and 2.2% in men. Among specific disorders, AN demonstrated a lifetime prevalence of 1.4% for women (range: 0.1–3.6%) and 0.2% for men (range: 0–0.3%), while BN showed prevalence rates of 1.9% in women (range: 0.3–4.6%) and 0.6% in men (range: 0.1–1.3%). BED was reported in 2.8% of women (range: 0.6–5.8%) and 1.0% of men (range: 0.3–2.0%) [8].

Further research indicates that OSFED has the highest lifetime prevalence at 7.4%, followed by AN (3.6%), BN (2.1%), and BED (2%). These findings reveal the widespread impact of eating disorders, highlighting gender disparities and underscoring the need for further research to understand the unique prevalence patterns across different disorders and populations [9].

## 4. Challenges, Mortality, Comorbidities, and the Need for Comprehensive Care

Feeding and EDs are among the most difficult mental health conditions to treat due to several complex factors. Firstly, these disorders lack specific phenotypes, making it challenging to identify consistent diagnostic markers or treatment predictors. Secondly, the progression of EDs varies widely among patients, with no predictable course of illness. Thirdly, individuals with EDs often demonstrate limited or no insight into their condition, impeding their willingness to seek or accept help from mental health professionals. In some cases, they may actively resist treatment or manipulate the therapeutic process, which can lead to treatment failure. Furthermore, EDs profoundly affect physical health, often causing severe medical complications and emergencies. These challenges underscore the importance of adopting a holistic, multidisciplinary treatment approach as recommended by international guidelines [5,6].

Mortality rates linked to EDs are among the highest across all mental health conditions. A meta-analysis by Arcelus et al. (2011) identifies AN as the mental disorder with the highest mortality rate. Similarly, Franko et al. (2013) found that AN carries a mortality rate of 4.37, while BN has a rate of 2.33. The risk of premature death tends to increase during the first decade of illness and is significantly influenced by patient age, with mortality rates rising from 3.2 in individuals aged 0–15 years to 6.6 in those aged 15–30 years. Additional factors associated with higher mortality rates include prolonged illness duration, alcohol or substance abuse, extremely low body mass index (BMI), suicidality (affecting 1 in 5 patients), and poor psychosocial functioning [10,11].

As depicted in Figure 1, eating disorders are frequently associated with numerous psychiatric comorbidities. The most prevalent comorbidities are affective disorders, including depression and bipolar disorder, along with anxiety disorders, obsessive–compulsive disorder, personality disorders, attention deficit hyperactivity disorder (ADHD), substance abuse, and suicidal ideation. These psychiatric comorbidities further complicate diagnosis and treatment, requiring tailored therapeutic interventions [12,13,14].

In addition to psychiatric comorbidities, Figure 1 illustrates the extensive physical complications caused by EDs, which impact multiple systems of the human body. These complications, driven by the effects of malnutrition and disordered eating behaviors, exacerbate the bidirectional relationship between EDs and physical health issues, creating a vicious cycle that hinders recovery. The complexity of these interactions highlights the necessity for comprehensive treatment approaches that consider the psychological, physiological, and systemic dimensions of EDs to improve outcomes and reduce mortality [12,13,14].

## 5. Medical Complications of Eating Disorders

EDs have far-reaching medical complications that affect multiple body systems. Cardiovascular issues, such as hypotension, bradycardia, QTc prolongation, and structural changes in the heart, increase the risk of severe outcomes, including heart failure [15]. Gastrointestinal complications range from tooth erosion and reflux to gastric dilation and esophageal rupture [15,16]. Hematologic abnormalities, including anemia, leukopenia, and bone marrow hypoplasia, are common, alongside significant impacts on bone health, such as osteopenia and osteoporosis, particularly in early-onset EDs [17,18,19]. Electrolyte imbalances like hypokalemia and metabolic disturbances, combined with sleep disorders and systemic impacts on the endocrine and neurological systems, highlight the pervasive health risks of EDs, necessitating comprehensive medical management [20,21,22]. A detailed table (Table 2) of these complications is provided below for reference.

## 6. Malnutrition

Malnutrition is a fundamental characteristic of EDs, as reflected in their diagnostic criteria and associated physiological consequences. Extensive research has investigated the impact of malnutrition across various ED types, revealing significant micronutrient and macronutrient deficiencies.

A 2015 meta-analysis found that individuals with pica had a 2.4-fold increased likelihood of experiencing anemia compared to the general population. This finding positions pica as a notable risk factor for anemia, comparable to well-established causes such as deficiencies in iron, vitamin B12, or folate, particularly among American women [23].

Specific nutrient deficiencies have been extensively studied in individuals with AN. A 2017 study involving 153 participants reported that 50% of individuals with AN had at least one micronutrient deficiency, with additional deficiencies often emerging during the refeeding process. Selenium was identified as the most frequently deficient trace element, while vitamins A and B9 were the most commonly deficient vitamins [24]. A larger study conducted in 2019 with 374 participants further corroborated these findings, noting that zinc deficiency was the most common (64.3%), followed by deficiencies in vitamin D (52.4%), copper (37.2%), selenium (20.5%), and vitamins B1 (15%), B9 (8.9%), and B12 (4.7%). The study also observed subtype-specific variations in nutrient deficiencies, with the binge–purging subtype exhibiting lower levels of selenium and vitamin B12 and the restricting subtype showing lower levels of copper [2].

In BED, nutrient intake patterns diverge from those observed in restrictive EDs. A 2022 study evaluating energy and nutrient intake in individuals with BED found that daily energy consumption exceeded recommended levels in women while remaining within normal limits for men. Despite the higher overall caloric intake, macronutrient profiles revealed excessive consumption of saturated fats and insufficient intake of omega-3 fatty acids. Both sexes showed deficiencies in vitamin D, selenium, and salt, with women also demonstrating higher rates of iron and folate deficiencies [25].

These findings highlight the pervasive and varied impact of malnutrition in EDs, with specific nutrient deficiencies contributing to systemic manifestations. Early identification and targeted correction of these deficiencies are critical to mitigating their physical and psychological consequences, ultimately supporting recovery and overall health.

## 7. Minerals: Dietary Sources, Functions, and Dermatological Impacts of Deficiency

The dietary sources, physiological functions, and dermatological impacts of key minerals are discussed below. Figure 2 illustrates the skin manifestations associated with deficiencies in copper, selenium, calcium, iron, and zinc, highlighting their importance as vital diagnostic markers, particularly in individuals with EDs.

### 7.1. Copper (Cu)

Copper is a vital trace element essential for maintaining overall physiological functions and skin health, particularly in individuals with EDs. The recommended daily intake for adults ranges from 1.1 to 2 mg/day, though absorption efficiency can vary significantly, ranging from 20% to 50%. Copper is most concentrated in the brain, liver, and kidneys, and its absorption is a tightly regulated process involving specific mechanisms in the gastrointestinal tract. In this process, dietary copper in the Cu^2^⁺ state is reduced to the Cu⁺ state before being absorbed. Once in the liver, copper is integrated into ceruloplasmin, released into the bloodstream, and distributed to tissues. Most of the absorbed copper is excreted through bile. Dietary sources of copper include beef liver, oysters, dark chocolate, salmon, pasta, mushrooms, nuts, and sunflower seeds [26].

Copper plays a crucial role in enzymatic activities through its involvement in cuproenzymes, which facilitate processes such as cellular respiration and gene regulation. In the skin, copper contributes to antioxidant defense, melanin synthesis, and extracellular matrix cross-linking, all of which are essential for pigmentation and wound healing. Copper deficiencies, often associated with the poor dietary intake typical of EDs, adversely affect skin health and other physiological systems. Dermatological signs of copper deficiency include hypopigmentation of the skin and hair, alopecia, and structural hair abnormalities such as pili torti, which presents as twisted or wiry strands. These symptoms are linked to reduced tyrosinase activity, a copper-dependent enzyme critical for melanin production [27].

Additional manifestations of copper deficiency include delayed wound healing, persistent fatigue, depression, and peripheral neuropathy. Severe cases, such as those seen in Menkes disease, highlight the serious implications of copper imbalance, including a wide spectrum of neurological and physical complications. In the context of EDs, addressing copper deficiency requires careful monitoring and supplementation, often in conjunction with zinc to ensure balanced nutrient restoration. This approach helps mitigate both dermatologic and systemic effects of copper deficiency, supporting recovery and overall health [27].

### 7.2. Selenium (Se)

Selenium is an essential trace element crucial for various cellular functions, including antioxidant protection, thyroid activity, and immune system support. Its primary dietary sources include poultry, grains, fish, vegetables, red meat, eggs, and selenium-enriched yeast. [28] After ingestion, selenium is absorbed in the gastrointestinal tract, processed in the liver, and distributed throughout the body. A significant portion of dietary selenium is converted into selenocysteine (Sec), an amino acid that integrates into selenoproteins critical for cellular functions. Absorption predominantly takes place in the small intestine, with selenium being excreted mainly via the urinary system [29]. The RDI for selenium is 55 μg/day for adults, increasing to 60 μg/day during pregnancy and 70 μg/day during lactation [30].

Selenium deficiency can manifest in a variety of dermatological and systemic symptoms. A 2005 case report highlighted that selenium deficiency may result in xerotic (dry) skin, irregular erythematous lesions, mild fissuring of the lips, and sparse, light-colored hair [31]. Additionally, selenium has been linked to mental health, with a comprehensive review in 2023 associating low selenium levels with depression and an increased risk of psychiatric disorders [32].

Given its critical role in cellular function and the risk of deficiency in people with eating disorders, monitoring the selenium status is essential. Supplementation may help mitigate symptoms such as skin dryness, hair thinning, and potential psychological effects, ultimately contributing to improved systemic health and recovery in individuals affected by malnutrition.

### 7.3. Calcium (Ca)

Calcium is a vital nutrient essential for maintaining strong bones and supporting numerous physiological processes. The recommended daily intake for an adult is approximately 1000 mg, which is necessary for skeletal integrity and overall bodily functions [30]. Calcium plays a key role in cellular signaling, including processes like survival and apoptosis, and its concentration in blood plasma is tightly regulated by three hormones: parathyroid hormone (PTH), calcitonin, and the active form of vitamin D3. Over 99% of the body’s calcium is stored in bones and teeth as hydroxyapatite, providing structural strength, while the remainder is distributed in soft tissues and bodily fluids. Calcium absorption occurs in the gastrointestinal tract in its ionized form via two primary mechanisms: transcellular active transport and paracellular passive transport. Dairy products are the main dietary source of calcium for many populations, but non-dairy foods like kale, spinach, broccoli, soy products, and beans also contribute, though their calcium bioavailability is typically lower [33].

Calcium’s role extends beyond structural support, playing a critical part in cellular signaling processes, including neurotransmission. Through these mechanisms, calcium influences thyroid and parathyroid function, often in conjunction with magnesium. This intricate interplay highlights calcium’s significant impact on mental health, with deficiencies linked to cognitive impairment, psychosis, anxiety, and, most notably, depression [34,35].

Dermatological manifestations of calcium deficiency are common, often affecting the quality of skin, hair, and nails, leading to dryness and thinning.

### 7.4. Iron (Fe)

Iron deficiency is one of the most common nutritional deficiencies observed in individuals with EDs, often exacerbated by red blood cell hemolysis, particularly in those who engage in intense physical activity. This deficiency can sometimes be masked by amenorrhea, a common symptom in EDs. DRI for iron differs by sex and age, with males requiring 8 mg/day and females 18 mg/day until the age of 50 [30].

Dietary iron is available in two forms: heme iron, which is derived from animal sources such as meat, fish, and poultry, and non-heme iron, found in plant-based foods like fruits, vegetables, and fortified products. Heme iron has a bioavailability roughly twice that of non-heme iron, but the absorption of both forms can be significantly enhanced by consuming vitamin C alongside iron-rich foods. Iron absorption primarily takes place in the duodenum, with excess iron stored in the spleen, bone marrow, and liver as ferritin or hemosiderin, or in muscles as myoglobin. Excretion occurs through feces, urine, the gastrointestinal tract, skin shedding, and blood loss [27].

Common symptoms of iron deficiency include persistent fatigue and weakness and difficulty maintaining body warmth. Dermatological signs are also common, with notable nail abnormalities such as koilonychia (spoon-shaped nails), which is characterized by an upward eversion of the nail plates and observed in about 5% of cases of iron deficiency anemia. Other mucocutaneous manifestations include angular cheilitis, pruritus, glossitis, and telogen effluvium (a form of hair loss). While the association between iron deficiency and hair loss remains a subject of debate, its role in various skin and mucosal changes is well documented [3,4].

Addressing iron deficiency in individuals with EDs is crucial for improving systemic health and mitigating dermatologic symptoms, emphasizing the importance of adequate dietary intake and, when necessary, supplementation.

### 7.5. Zinc (Zn)

Zinc is an essential trace element crucial for numerous physiological functions, particularly in the skin, which is the third most zinc-abundant tissue in the body, following skeletal muscle (60%) and bones (30%). Around 5% of the body’s total zinc is concentrated in the liver and skin. In the skin, zinc levels are higher in the epidermis compared to the dermis, where it plays a vital role in various cellular activities. Zinc is a structural component of zinc finger motifs found in DNA- and RNA-binding proteins, with approximately 10% of human proteins requiring zinc binding. It supports the functions of multiple skin cell types, including keratinocytes, Langerhans cells, melanocytes, mast cells, T-cells, dendritic cells, adipocytes, fibroblasts, and endothelial cells [36].

The RDI for zinc is 11 mg for adult males and 8 mg for adult females. Zinc is abundant in foods such as fish, beef, oysters, soybeans, liver, and beans [30]. Its absorption mainly takes place in the jejunum and duodenum, with smaller amounts absorbed in the ileum and large intestine. Once absorbed, zinc binds to albumin for transport to the liver and is predominantly excreted via the gastrointestinal tract [27].

Assessing zinc status can be challenging in patients with significant weight loss, as tissue catabolism can release zinc, temporarily elevating serum levels and skewing results. Zinc supplementation can benefit patients with even mild deficiencies or inadequate dietary intake [3].

Zinc deficiency can lead to various clinical manifestations. Acrodermatitis enteropathica (AE) is a rare autosomal recessive disorder caused by mutations in genes responsible for intestinal zinc transport. This condition leads to impaired zinc absorption and is characterized by a triad of periorificial and acral dermatitis, diarrhea, and alopecia, though this full presentation occurs in only about 20% of cases. AE’s dermatologic signs often include vesicular or pustular rashes resembling psoriasiform dermatitis, affecting areas like the feet, knees, and hands [4,27]. Other conditions, such as epidermodysplasia verruciformis, are linked to zinc transporter mutations and an increased risk of non-melanoma skin cancers [36].

Acquired zinc deficiency manifests as alopecia, dermatitis (particularly in the perioral, perineal, and acral regions), hypogeusia (reduced taste), and anorexia [3,4]. Zinc deficiency also impairs weight gain in underweight individuals by hindering cellular growth and muscle mass accumulation. Additionally, zinc is crucial for vitamin A metabolism, and its deficiency can lead to elevated serum carotenoid levels [3].

From a mental health perspective, zinc deficiency may exacerbate depressive symptoms due to its role in GABA neurotransmission, highlighting the importance of maintaining adequate zinc levels to support both physical and psychological health [3].

## 8. Vitamins: Dietary Sources, Functions, and Dermatological Impacts of Deficiency

The dietary sources, physiological functions, and dermatological impacts of key vitamins are discussed below. Figure 3 illustrates the skin manifestations associated with deficiencies in vitamins A, B2, B3, B6, B9, B12, C, and D, highlighting their importance as vital diagnostic markers, particularly in individuals with ED.

### 8.1. Vitamin A (Retinol)

Vitamin A is an essential fat-soluble vitamin obtained from both animal- and plant-based foods. Rich dietary sources include animal products like poultry and beef, as well as plant-based options such as pumpkin, sweet potatoes, carrots, spinach, and butternut squash. Its absorption occurs in the small intestine, facilitated by pancreatic enzymes and bile products. The RDI of retinol is approximately 900 μg for adult males and 700 μg for adult females [30].

Vitamin A plays a pivotal role in optical health as a key component of rhodopsin, a protein necessary for vision, particularly in low-light conditions. It is also vital for maintaining the integrity and function of the conjunctiva and cornea. Deficiency may result in various symptoms, involving impaired night vision, xerophthalmia, and Bitot spots—grey-white plaques on the conjunctiva. Severe deficiencies, if untreated, may result in irreversible blindness [27]. Additionally, vitamin A has been linked to mental health, with several studies associating retinol levels with depression [32].

Excessive vitamin A intake, or hypervitaminosis A, can cause carotenoderma, a condition characterized by yellow-orange skin pigmentation due to the excretion and accumulation of carotenes in the stratum corneum through sebaceous and eccrine glands. Other manifestations of hypervitaminosis A include cheilitis, stomatitis, skin desquamation, pruritus, and alopecia [4].

Conversely, hypovitaminosis A is linked to dermatological conditions such as xerosis (dry skin) and phrynoderma, colloquially known as “toad skin”. Phrynoderma is characterized by follicular, hyperkeratotic papules, most commonly found on the extremities and buttocks but also appearing on the shoulders, back, posterior neck, and abdomen. These papules are usually asymptomatic [4].

Vitamin A’s multifaceted role underscores its importance in maintaining vision, skin health, and overall physiological well-being, while imbalances in its levels—whether excessive or deficient—can have significant systemic and dermatological consequences.

### 8.2. Vitamin B1 (Thiamin)

Thiamin deficiency is prevalent among individuals with EDs, particularly those who abuse alcohol or severely limit carbohydrate intake. The RDI of thiamin is 1.2 mg for adult males and 1.1 mg for adult females [30]. Good dietary sources include whole grains, legumes, pork, fish, nuts, seeds, and fortified foods.

Thiamin deficiency is primarily associated with neuropsychiatric and neurological conditions, including Korsakoff syndrome, which is characterized by dementia, nystagmus, ophthalmoplegia, ataxia, and peripheral neuropathy. Additionally, thiamin deficiency may exacerbate psychiatric disorders, such as depression, and contribute to cognitive impairment. The importance of thiamin in cellular energy metabolism highlights its role in maintaining both neurological and psychological health [3].

### 8.3. Vitamin B2 (Riboflavin)

Riboflavin deficiency is frequently observed in individuals with EDs, especially those with excessive alcohol consumption. The RDA for riboflavin is 1.3 mg per day for adult men and 1.1 mg per day for non-pregnant adult women [30]. Significant dietary sources of riboflavin include fortified cereals, leafy green vegetables, poultry, milk, meat, eggs, and almonds. Riboflavin is absorbed in the proximal small intestine and metabolized in the liver, where it is converted into flavoproteins that play a crucial role in various cellular processes [27].

Notably, riboflavin levels can be reduced in individuals using oral contraceptives, which is relevant because many patients with EDs experience amenorrhea and may be prescribed such medications [3]. Riboflavin deficiency manifests in various mucosal and skin-related symptoms, including pharyngitis, cheilitis, glossitis, stomatitis, and seborrheic dermatitis. Acute deficiency may result in severe symptoms such as mucositis, epidermal necrolysis, and deep-red erythema, while chronic deficiency can cause oral–ocular–genital syndrome. Oral symptoms include cheilitis, angular stomatitis with papules, and glossitis. Ocular symptoms may present as conjunctivitis and photophobia, and genital symptoms, particularly in men, include pruritic scrotal dermatitis [4].

Both thiamin and riboflavin deficiencies emphasize the significant impact of nutrient imbalances on neurological, psychiatric, and dermatological health, emphasizing the need for early detection and appropriate nutritional support in individuals with EDs [3,4].

### 8.4. Vitamin B3 (Niacin)

Niacin, a water-soluble vitamin, is found abundantly in animal-derived foods, legumes, nuts, and grains. It can also be synthesized in the body from tryptophan, a process requiring adequate dietary intake of foods such as poultry, seeds, nuts, soybeans, legumes, and beef. To produce 1 mg of niacin, approximately 60 mg of dietary tryptophan is needed, with sufficient levels of vitamin B6 being a prerequisite for this conversion. Niacin is absorbed in the small intestine and metabolized in the liver, where tryptophan is converted into niacin [27]. The recommended daily allowance (RDA) for vitamin B3 in the United States is 16 mg per day for adult men and 14 mg per day for non-pregnant adult women [30].

Deficiency in niacin or tryptophan can result in pellagra, a condition classically characterized by the “4 Ds”: dermatitis, diarrhea, dementia, and, in severe cases, death. The hallmark dermatological symptom of pellagra is a photosensitive rash that initially appears as erythema and edema following sun exposure. This rash might be painful, cause burning sensations, or induce pruritus, often mimicking sunburn. In more severe cases, vesicles or bullae can develop, referred to as “wet pellagra”. Frequently affected areas include the dorsal surfaces of the hands (up to 97% of cases), a butterfly-shaped pattern on the face, and a broad band across the neck and chest, commonly known as Casal’s necklace [4].

Variants of cutaneous pellagra include the following:

Perineal, genital, and mucosal erosions, often associated with atrophic glossitis, cheilitis, and vaginitis.

Hyperkeratosis and hyperpigmentation, typically symmetrical and found over bony prominences such as the knees and elbows, developing gradually.

Sebaceous gland prominence, resembling seborrheic dermatitis [37].

These diverse manifestations highlight the significant impact of niacin deficiency on the skin and systemic health.

### 8.5. Vitamin B6 (Pyridoxine)

Vitamin B6 is present in high concentrations in poultry, fish, potatoes, bananas, fortified cereals, and chickpeas, with higher bioavailability in meat sources compared to plant sources. Pyridoxine is absorbed in the jejunum, metabolized in the liver, and excreted in the urine [27]. The RDA for vitamin B6 is 1.3 mg daily for men and women aged 14 to 50 years, increasing to 1.7 mg for men and 1.5 mg for women over 50. Deficiency is more prevalent in individuals with alcohol use disorders and can exacerbate eating disorders due to its role in serotonin metabolism and appetite regulation [30].

Pyridoxine deficiency presents with a variety of dermatological and systemic symptoms. Skin-related manifestations include stomatitis, glossitis, cheilitis, and, in severe cases, seborrheic dermatitis. The most frequent dermatological sign is a seborrheic-like rash, typically localized to the scalp, face, neck, shoulders, perineum, and buttocks. In rare cases, pyridoxine deficiency can result in a pellagra-like dermatitis on the dorsal surfaces of the extremities. Systemically, it can cause microcytic anemia, peripheral neuropathy, seizures, and depression [4,27].

Both niacin and pyridoxine are critical for maintaining skin integrity, neurological function, and mental health. Deficiencies in these vitamins highlight the importance of adequate dietary intake and monitoring, particularly in individuals with eating disorders, where nutritional imbalances are common and often severe [4,27].

### 8.6. Vitamin B12 (Cobalamin)

Vitamin B12 is a critical nutrient essential for neurological health, hematopoiesis, and DNA synthesis. The RDA is 2.4 μg per day for individuals over 14 years of age [30]. Rich dietary sources of cobalamin include animal-based foods such as liver, clams, fish (e.g., trout and salmon), meat, poultry, eggs, and dairy products, as well as fortified plant-based alternatives like specific cereals and nutritional yeast. Cobalamin is taken up in the distal ileum after binding to intrinsic factor, a glycoprotein produced by gastric parietal cells. Elevated plasma homocysteine levels may serve as an indicator of deficiency [27].

A hallmark but nonspecific sign of vitamin B12 deficiency is Hunter’s glossitis, which is characterized by diffuse erythema and atrophy of lingual papillae, frequently involving over half the tongue. This condition occurs in approximately 25% of cases. More specific early signs, which can precede macrocytic anemia, include atrophic linear lesions on the tongue and hard palate. Symptoms of glossitis typically involve a sore mouth, burning sensations, and altered taste. Additional oral manifestations may include angular cheilitis and aphthous stomatitis.

Cutaneous signs of vitamin B12 deficiency can present as hyperpigmentation, resembling Addison’s disease, and longitudinal streaks on nails. Skin depigmentation, such as vitiligo, may also occur. These mucocutaneous symptoms generally resolve with prompt B12 supplementation, highlighting the importance of early diagnosis to prevent severe complications. Left untreated, vitamin B12 deficiency can result in megaloblastic anemia and irreversible neuropsychiatric disorders [4].

### 8.7. Vitamin C (Ascorbic Acid)

Vitamin C is a water-soluble vitamin fundamental for collagen synthesis, antioxidant defense, and immune support. The RDA is 90 mg per day for men and 75 mg per day for women aged 18 and older, with increased requirements during pregnancy, lactation, and for individuals who smoke [30]. Rich dietary sources include citrus fruits (e.g., lemons, grapefruits, and oranges), guava, tomatoes, strawberries, spinach, bell peppers, kiwi, broccoli, and Brussels sprouts. Absorption occurs in the distal small intestine, and vitamin C is excreted primarily via the kidneys [27]. Urinary excretion testing provides a more accurate measure of tissue saturation compared to blood tests, as it reflects the body’s storage status rather than recent intake fluctuations. In individuals with eating disorders (EDs), the combination of inadequate dietary intake and intense physical activity significantly elevates the risk of deficiency [3].

Severe and sustained vitamin C deficiency results in scurvy, which can develop within 1 to 3 months of depletion. Scurvy is characterized by the “four Hs”: hemorrhagic signs, hyperkeratosis of hair follicles, hypochondriasis, and hematologic abnormalities such as anemia. Early symptoms include systemic fatigue, malaise, musculoskeletal pain (arthralgias), anorexia, weight loss, diarrhea, sideropenic anemia (iron deficiency anemia due to bleeding), and emotional instability. Advanced symptoms include easy bleeding, bruising, and impaired wound healing. In patients with poor dental health, scurvy often manifests as hemorrhagic gingivitis, presenting with red, spongy gums that may progress to necrosis and tooth loss, leaving the area prone to infection [4].

If untreated, scurvy can lead to severe complications, including death caused by profound bleeding or infections. These serious outcomes point out the importance of recognizing and addressing vitamin C deficiency early, particularly in populations at higher risk, such as individuals with EDs [4].

### 8.8. Vitamin D (Calciferol)

Vitamin D is a fat-soluble vitamin essential for calcium and phosphate homeostasis, bone health, and immune function. Abundant dietary sources include fatty fish (e.g., sardines, mackerel, and salmon), egg yolks, cod liver oil, fortified foods (e.g., cereals, milk, and orange juice), and UV-exposed mushrooms. The RDA is 15 μg daily for adults aged under 70 years and 20 μg daily for those over 70 [30]. Vitamin D is synthesized in the skin as vitamin D3 upon exposure to UV radiation. Dietary forms of vitamin D, including D2 and D3, are metabolized in the liver to 25-hydroxyvitamin D, the primary circulating form. In the kidneys, under the regulation of parathyroid hormone (PTH), 25-hydroxyvitamin D is converted to its active form, calcitriol, which regulates calcium and phosphate absorption in the intestines, mobilization in bones, and reabsorption in the kidneys [38].

Deficiency in vitamin D often manifests dermatologically as alopecia and dry, flaky skin prone to infections, UV damage, and exacerbation of inflammatory skin conditions [4]. Beyond physical symptoms, vitamin D deficiency has been associated with mental health disturbances. Studies have shown that individuals with depression often exhibit significantly lower serum vitamin D levels, suggesting a link between calciferol deficiency and mood instability, increased irritability, and other neuropsychiatric symptoms [32].

Vitamin D’s multifaceted role in maintaining both physical and mental health underscores the importance of adequate dietary intake, skin synthesis through sunlight exposure, and supplementation when necessary, particularly in populations at risk of deficiency.

In summary, Table 3 below provides a comprehensive overview of common nutritional deficiencies associated with eating disorders, highlighting their corresponding dermatological and mental health manifestations.

## 9. Malnutrition Due to Behavioral Patterns and Their Dermatologic Manifestations

Malnutrition resulting from disordered eating behaviors is closely associated with a range of dermatologic manifestations. In 1987, Gupta and colleagues [39] were the first to systematically categorize skin changes linked to eating behaviors, dividing them into four distinct groups: (a) manifestations due to starvation or malnutrition, such as lanugo-like body hair, dry skin, brittle nails, and carotenoderma; (b) effects of self-induced vomiting, including hand calluses (Russell’s sign), dental enamel erosion, gingivitis, and symptoms mimicking Sjögren’s syndrome; (c) dermatologic side effects from the use of laxatives, diuretics, or emetics; and (d) skin conditions related to other psychiatric comorbidities, such as hand dermatitis resulting from compulsive washing [39].

Building on this foundational work, Glorio et al. [40] conducted a comprehensive observational, cross-sectional, and prospective study in 2000 to further identify dermatologic “guiding signs” indicative of EDs. Their research highlighted specific markers, including Russell’s sign (calluses on the dorsal aspects of the hands from repetitive self-induced vomiting), hypertrichosis (fine, lanugo-like body hair), significant dental enamel erosion (perimylolysis), and self-inflicted dermatitis. These findings have significantly enhanced the ability of clinicians to recognize dermatological indicators of EDs, facilitating early diagnosis and intervention [40].

Figure 4 below presents a schematic representation of behavioral patterns associated with EDs.

### 9.1. Dermatologic Manifestations Due to Starvation

#### 9.1.1. Lanugo-like Hair Growth

Lanugo-like hair appears as fine, soft, and lightly pigmented strands that commonly develop on the back, abdomen, and forearms. In AN, particularly among younger patients, lanugo may also emerge on the face, neck, and arms. This hair growth serves as a protective layer, helping the body conserve warmth [41]. Unlike hair changes associated with hormonal imbalances, lanugo-like hair in AN is not indicative of virilization. Instead, it is linked to reduced activity of the 5-alpha-reductase enzyme system, likely influenced by hypothyroidism [22]. While this condition is a hallmark of AN and rare in other malnutrition states, it significantly improves with nutritional rehabilitation, leading to a gradual reduction in lanugo as the body recovers [42].

#### 9.1.2. Xerosis (Dry Skin)

Xerosis, or dry skin, also known as asteatosis, is commonly caused by dehydration resulting from behaviors like purging, excessive use of laxatives and diuretics, and repetitive washing, all of which are frequently observed in AN. Prolonged starvation depletes calories and vitamins, reducing sebaceous gland activity and exacerbating dryness. Symptoms may include dry mouth, thinning facial features, and electrolyte imbalances. Sebum production typically begins to decrease within 1–4 weeks of starvation. Lipid changes include reductions in triglycerides, wax esters, and cholesterol levels, while squalene levels often remain unaffected. While emollients may provide temporary relief, complete resolution of xerosis requires adequate nutritional recovery [43,44].

#### 9.1.3. Acrocyanosis

This condition is characterized by a bluish discoloration and cold sensations in the hands and feet, commonly seen in more severe cases of AN. It is caused by narrowed arterioles and dilated veins in the skin and is often accompanied by pallor, reduced pulse rate, and elevated fasting glucose levels [43]. Particularly prevalent in adolescent females, acrocyanosis is believed to be a heat-conserving response to prevent excessive heat loss or dehydration. Related vascular conditions, such as Raynaud’s phenomenon and chilblains (perniosis), which present as reddish to purplish patches on extremities, may also occur. Fortunately, acrocyanosis is reversible and typically resolves with proper nutritional rehabilitation and weight restoration [41].

#### 9.1.4. Pruritus (Itching)

Pruritus, or persistent itching, is commonly reported in individuals with AN, particularly at lower body weights. Studies suggest that up to 58% of patients experience pruritus in malnourished states, though the condition often eases as weight is regained and BMI improves [45]. Severe cases may lead to excessive scratching or compulsive washing [46]. While some instances are linked to xerosis due to malnutrition, pruritus can also occur independently, with no underlying liver, kidney, or androgen-related conditions. Hypothyroidism, a possible result of malnutrition, might contribute in certain cases [45]. Potential causes include psychological factors, altered sensory perception, hormonal imbalances, and changes in thermoregulation or regional blood flow [47]. Eczema, observed in nearly half of AN patients, may also contribute but often resolves with nutritional recovery [45]. Additionally, hormonal imbalances common in AN, such as decreased serum androgens and lower sebaceous gland function from restricted caloric intake, might play a role [43].

#### 9.1.5. Purpura

Purpura appears as diffuse erythematous lesions caused by thrombocytopenia, secondary to bone marrow suppression. Nearly 50% of individuals with AN exhibit bone marrow atrophy, sometimes accompanied by gelatinous marrow transformation. Peripheral blood cell count alterations, while common, are not reliable indicators of the severity of marrow atrophy. Rarely, diffuse reticulate purpura presents as a purpuric rash, primarily on the trunk, that subsides rapidly with nutritional rehabilitation [18]. Increased capillary fragility and reduced dermal support from malnutrition further predispose individuals to purpura, which resolves with weight gain and improved nutrition [48]. A rare variant, diffuse reticulate purpura, has been documented in AN patients with severe malnutrition, presenting as a purpuric reticulate rash predominantly on the trunk that subsides rapidly with refeeding [49].

#### 9.1.6. Nail Abnormalities

Nail changes, such as periungual erythema (20–48%), fragility (15–33%), longitudinal striae (15%), and onychoschizia (9%), are frequently observed in AN [43]. These abnormalities are often linked to nutritional deficiencies or compulsive behaviors [46]. Nail fragility results from severe dryness and lipid depletion, while koilonychia (spoon-shaped nails) is typically associated with iron deficiency. Additional findings may include Beau’s lines, Terry’s nails, Muehrcke’s lines, and color changes due to B12 deficiency or splinter hemorrhages. These symptoms often resolve as nutritional health improves [50].

#### 9.1.7. Oral Cavity Manifestations

Cheilitis, angular cheilitis, geographic tongue, and aphthous ulcers are common among individuals experiencing severe malnutrition or starvation [51]. Angular cheilitis is frequently associated with deficiencies in vitamins such as riboflavin, while gingival bleeding may indicate thrombocytopenia [42]. Reduced salivary flow (hyposalivation) is another frequent finding in malnourished patients, exacerbating oral symptoms [52].

#### 9.1.8. Hair Loss and Hair Abnormalities

Hair health is a key clinical indicator of nutritional status in eating disorders. Studies show that hair issues, particularly hair loss or effluvium, are more common in the bulimic subtype of AN than in the restrictive type [46]. A range of hair conditions, such as alopecia, brittle or opaque hair, and diffuse hair thinning, affects 17–61% of individuals with eating disorders. Telogen effluvium, characterized by an increase in resting-phase hairs, is a hallmark of malnutrition-induced hair cycle disruption. These hair changes reflect internal protein depletion and other deficiencies caused by crash diets [43].

Clinical observations highlight how hair loss can point out the severity of eating disorders and motivate recovery. For example, Kashif et al. [53] reported cases where visible hair thinning prompted individuals to reassess their eating habits and seek help. An 18-year-old with anorexia nervosa recognized the toll of her condition through noticeable hair loss, leading her to pursue recovery. Similarly, a 35-year-old woman experiencing hair loss from laxative abuse became aware of her disorder’s impact. These cases emphasize the need for healthcare providers to consider eating disorders when addressing unexplained hair loss, reinforcing its value as a diagnostic clue [53].

### 9.2. Dermatologic Manifestations Due to Self-Induced Vomiting

#### 9.2.1. Russell’s Sign

A hallmark indicator of purging in AN, Russell’s sign refers to calluses on the knuckles of the dominant hand caused by repeated contact with teeth during self-induced vomiting. This sign is particularly prevalent in individuals with the purging subtype of AN and serves as an important clinical clue for dermatologists and other healthcare providers in identifying undiagnosed eating disorders. The calluses, formed by repetitive friction, often display histological features such as epidermal hyperplasia and dermal fibrosis, reflecting the chronicity of the behavior [54].

#### 9.2.2. Herpetic Whitlow and Jehany Sign

Painful skin infection caused by the herpes simplex virus (HSV), typically affecting the fingertips (distal phalanx) and occasionally the toes. In younger children, the infection often results from autoinoculation with HSV-1, whereas in adolescents and adults between 20 and 30 years, it is more frequently associated with HSV-2. Early symptoms, or prodromal signs, may include sensations of burning, itching, or tingling in the affected finger or even the entire limb, usually followed by redness, pain, and development of vesicles on the skin. In 2023 the term “Jehany Sign” has been proposed from Aljehani et al., in order to describe the appearance of recurrent, painful pustules that erode and dry on the plantar surface of the dominant hand’s finger. This condition is associated with prolonged self-induced vomiting. It develops due to repetitive insertion of the finger into the oral cavity, exposing it to vomit and oral secretions while stimulating the gag reflex at the back of the throat. Notably, this phenomenon occurs independently of prior exposure to the herpes simplex virus [55].

#### 9.2.3. Oral and Salivary Gland Manifestations

The physical and chemical stresses of self-induced vomiting, combined with poor nutritional status, contribute to significant oral changes. Dental enamel erosion is one of the most recognizable features, caused by frequent exposure to stomach acid. Additionally, bilateral enlargement of the parotid glands, known as sialadenosis, is a common finding in bulimia nervosa. Sialadenosis is a noninflammatory, painless swelling of the salivary glands, likely resulting from chronic vomiting and altered salivary gland function [56].

#### 9.2.4. Other

Purging behaviors can lead to additional dermatologic findings, including petechiae (tiny red or purple spots) around the eyes, caused by the increased pressure from vomiting. In BN, subconjunctival hemorrhages (bleeding beneath the surface of the eye) may occur due to the physical strain associated with frequent vomiting [57].

### 9.3. Dermatologic Manifestations Due to Substance Abuse

Substance abuse associated with EDs often exacerbates the systemic and dermatologic complications seen in conditions like purging-type AN and BN. The misuse of laxatives, diuretics, appetite suppressants, and emetics leads to a wide range of skin and systemic effects that provide critical diagnostic insights [43].

#### 9.3.1. Reactions to Laxatives and Diuretics

Chronic laxative use, particularly laxatives containing phenolphthalein, can lead to urticaria (hives), while diuretics such as furosemide may cause lichenoid eruptions, pruritus, and even bullous lesions. Prolonged use of laxatives containing senna has been linked to finger clubbing (hypertrophic osteoarthropathy), a condition that is reversible upon cessation. Dehydration caused by the misuse of diuretics and laxatives worsens xerosis (dry skin) and may also contribute to fluid retention and edema, complicating the physical presentation of eating disorders [41,58].

#### 9.3.2. Effects of Emetic Abuse

The misuse of emetics, such as ipecac syrup, poses significant risks, both dermatologic and systemic. Chronic ipecac use can cause skin changes that resemble dermatomyositis (violet or dusky red rash typically on the face, eyelids, knuckles, elbows, knees, chest, or back, which may be itchy or painful), in addition to serious systemic effects like myopathy (muscle weakness) and myocarditis (inflammation of the heart muscle). These manifestations highlight the harmful and far-reaching consequences of emetic abuse in ED patients [59].

#### 9.3.3. Acidic Food Consumption and Dental Erosion

Individuals with eating disorders often consume acidic, low-calorie “slimming” foods, such as citrus fruits and juices with a pH of around 3.5. These foods, when used as dietary staples, can accelerate enamel demineralization, especially in patients engaging in chronic restrictive eating. Self-induced vomiting, frequently associated with eating disorders, further exacerbates this problem by lowering salivary pH, increasing the risk of significant dental erosion over time [50,60].

#### 9.3.4. Carotenoderma

A distinctive dermatologic feature in restrictive eating disorders is carotenoderma, which manifests as a yellowish discoloration of the skin, particularly in thicker areas such as the palms, soles, and nasolabial folds. This condition results from hypercarotenaemia, caused by excessive intake of carotenoid-rich, low-calorie vegetables, a common dietary choice in restrictive AN and BN. Metabolic impairments in vitamin A and lipid processing, such as those caused by hypothyroidism, may further contribute to this condition. [41] Unlike other malnourished states where carotene levels are typically reduced, hypercarotenaemia is a unique marker of eating disorders and often resolves with nutritional recovery [43].

### 9.4. Dermatologic Manifestations Due to Psychiatric Conditions

EDs frequently present with distinct skin and behavioral manifestations, offering critical diagnostic insights into the physical and psychological challenges faced by individuals. Among these are self-inflicted injuries and trichotillomania (TTM), which reflect the multifaceted nature of EDs. Recognizing these signs is essential for early identification and holistic management of these complex conditions [50,61,62].

#### 9.4.1. Compulsive Behaviors

Compulsive behaviors such as nail-biting (onychophagia) and nail-picking (onychotillomania) are frequently observed in individuals with eating disorders, reflecting an overlap with obsessive–compulsive tendencies. These repetitive behaviors can exacerbate dermatologic symptoms and often serve as visible markers of underlying psychological distress [50].

#### 9.4.2. Self-Harming Behaviors

Self-inflicted injuries are a common finding among ED patients, often stemming from emotional distress, depression, and compulsive tendencies. Scars from cigarette burns, skin-picking, or excoriating acne are frequently observed in both AN and BN [63]. In some cases, individuals with eating disorders—particularly healthcare workers—may engage in self-phlebotomy, causing severe, unexplained anemia. Dermatologists should remain vigilant for needle marks, which could indicate this behavior, although discontinuation of purging behaviors can reduce related lesions [64]. These behaviors are believed to be a physical manifestation of psychological struggles and are particularly concerning due to their association with impulsivity and auto-aggressive tendencies [61].

#### 9.4.3. Trichotillomania (TTM)

TTM, characterized by compulsive hair-pulling, results in noticeable hair loss that frequently coexists with eating disorders. Both TTM and EDs are part of the obsessive–compulsive spectrum and share repetitive, compulsive behavioral patterns. Research indicates that individuals with body-focused repetitive actions, such as TTM or severe nail-biting, have a higher likelihood of developing EDs. This connection highlights shared psychological traits, including difficulty managing compulsions and heightened distress. Identifying and addressing TTM is critical to understanding the broader compulsive tendencies that may influence ED behaviors [62].

Table 4 below presents skin manifestations associated with malnutrition resulting from behavioral patterns, linking specific dermatological signs to their corresponding eating disorders.

## 10. Other Signs in Eating Disorders

### 10.1. Erythema Ab Igne (EAI)

EAI is a pigmented, net-like skin condition caused by extended exposure to low-level heat (typically between 43–47 °C) that does not cause burns. This condition often arises in individuals with anorexia nervosa (AN), who experience heightened cold sensitivity and reduced core body temperature, leading to excessive use of heat sources for comfort. AN patients frequently exhibit abnormal vascular responses to cold, such as reduced blood flow to extremities and increased vasoconstriction, further contributing to behaviors that result in EAI. Although EAI is benign and resolves after discontinuing heat exposure, it highlights the physiological adaptations and coping mechanisms employed by individuals with EDs [65].

### 10.2. Acne and Eating Disorders

Acne is a common concern in individuals with eating disorders, particularly AN, with nearly half of those affected experiencing acne. This condition is often exacerbated during phases of re-nutrition and weight restoration, primarily due to hormonal fluctuations and disruptions in the hypothalamic–pituitary–gonadal axis, along with broader endocrine imbalances characteristic of AN [43].

Interestingly, studies suggest a potential bidirectional relationship between acne and eating disorders. For some individuals, self-esteem issues stemming from acne may lead to the adoption of restrictive “acne diets”, which, although aimed at improving skin health, can inadvertently increase the risk of developing disordered eating behaviors. Conversely, the malnutrition and hormonal dysregulation seen in eating disorders may contribute to the onset or worsening of acne [66].

Given these dynamics, dermatologists play a crucial role in recognizing how acne can both influence and be influenced by the psychological and nutritional challenges associated with eating disorders. This awareness is essential for providing holistic care, as addressing the interplay between skin health, nutrition, and mental health can significantly improve patient outcomes [66].

### 10.3. Striae Distensae

Striae distensae (stretch marks) have been documented in 11.7% of patients with eating disorders, with a notable predominance in males diagnosed with AN. Morphologically, these striae appear identical to those seen in individuals without AN, though their presence in the context of severe weight fluctuations in AN warrants further investigation [42].

While the exact pathogenic mechanism remains unclear, hypercortisolemia—a condition commonly associated with stress and starvation—is believed to play a contributing role. The relationship between cortisol dysregulation and the occurrence of striae in AN patients remains an area of ongoing research [43].

Although there is no definitive treatment for striae distensae, limited application of topical tretinoin for up to three months may offer localized improvement in some cases. Additional research is required to clarify the underlying mechanisms and explore effective therapeutic options [42].

## 11. Gender-Specific Dermatologic and Systemic Presentation

### 11.1. Polycystic Ovarian Syndrome (PCOS) and Eating Disorders

PCOS and BN share overlapping hormonal and metabolic dysregulations, with dermatologic signs often providing insight into these conditions. [67] In PCOS, hyperandrogenism, characterized by elevated androgen levels, commonly results in acne, hirsutism (excessive facial and body hair growth), and androgenic alopecia (scalp hair thinning and loss). These visible symptoms of androgen excess can severely impact body image, leading to heightened anxiety and body dissatisfaction, which may increase vulnerability to disordered eating behaviors. For individuals with BN, binge eating is frequently driven by psychological stress and impaired impulse control, further complicating management [7].

### 11.2. Insulin Resistance and Dermatologic Signs

Insulin resistance, a hallmark of PCOS, often manifests as acanthosis nigricans, characterized by hyperpigmented, velvety patches of skin in areas such as the neck and axillae. This dermatologic marker indicates underlying metabolic dysregulation and can compound the challenges of PCOS. Insulin resistance may also drive intense carbohydrate cravings, predisposing individuals to overeating and binge-eating episodes. These interconnected hormonal, metabolic, and psychological factors accentuate the necessity for comprehensive treatment approaches that tackle both dermatologic and behavioral components [7].

### 11.3. AN in Males: Unique Considerations and Dermatologic Insights

AN in males is a multifaceted condition shaped by complex biopsychosocial factors, including societal pressures, occupational demands, and rigorous physical activity. Additionally, male-specific elements such as weight history, trauma, sexual orientation, depression, body image concerns, and media influence contribute to its development and progression [68]. Despite these distinctions, male AN is underrepresented in research, leading to significant gaps in understanding its etiology, progression, and optimal treatment strategies.

Males with AN frequently present with extremely low BMI, endocrine dysfunctions such as reduced testosterone levels, and systemic complications including anemia and liver dysfunction. These differences indicate the need for tailored diagnostic and therapeutic approaches to address the unique challenges faced by males with this condition [69].

The restrictive nature of male AN complicates diagnosis, as it often limits the manifestation of dermatological signs commonly associated with purging behaviors. For example, indicators like Russell’s sign or perimylolysis are less relevant due to the absence of vomiting or laxative abuse in the restrictive subtype. Similarly, dermatological features frequently seen in females with AN—such as lanugo-like hair, brittle nails, angular cheilitis, and self-inflicted dermatitis—are rare or less pronounced in males. Common female-associated signs like thinning hair, brittle texture, and acne are also infrequently reported in males [69].

This lack of overt dermatologic markers poses challenges for early detection and emphasizes the need for improved diagnostic criteria that consider the subtler presentation of AN in males. Compounding the issue, males with AN often have a worse prognosis compared to females, with restrictive eating patterns being the predominant subtype, further excluding behaviors like purging or laxative misuse [69].

## 12. Discussion

This review highlights the intricate relationship between nutritional deficiencies and dermatological manifestations in individuals with EDs, so it would be remiss not to mention the moderating factors of EDs, as they play a crucial role in shaping both the development and expression of eating disorders, interacting with genetic, biological, psychological, and sociocultural risk components. Genetic factors, including variations in neurotransmitters and hormones like dopamine and serotonin, significantly influence appetite regulation and reward processing across different ED subtypes. Sex-specific risks further highlight this relationship, with females exhibiting a greater genetic susceptibility compared to males. Biological influences, such as gut microbiota imbalances, also contribute to the risk of EDs, with prolonged satiety often observed in AN and impaired satiety in BED. Additionally, hormonal changes during key developmental stages, such as increased testosterone or cortisol exposure in utero, appear to heighten vulnerability to EDs, demonstrating the multifaceted nature of these moderating influences [70].

Furthermore, body image issues stand out as a critical psychological and sociocultural moderating factor in the onset and progression of EDs. High levels of dissatisfaction with one’s body and the internalization of unrealistic beauty standards are frequently linked to the development of EDs, while overemphasis on weight and shape often characterizes ongoing ED symptoms. Exposure to body-focused content in traditional (e.g., magazines and television) and social media has been shown to elevate ED risk among both males and females. Sociocultural influences, including the unique challenges faced by LGBTQ+ individuals and certain ethnic minorities, further exacerbate body image concerns. Activities that prioritize physical appearance and performance, such as competitive sports or contests, can also amplify these pressures, fostering disordered eating behaviors. Overall, body image plays a pivotal role as a moderating factor, shaping how individual vulnerabilities interact with societal influences to impact the risk of developing EDs [70].

The review also emphasizes the diagnostic significance of skin changes as reflections of systemic health. Alterations in the skin, hair, and nails serve as critical diagnostic markers, particularly in cases where patients may underreport or deny the severity of their condition [53]. Additionally, evidence consistently demonstrates that specific nutrient deficiencies correspond to distinct dermatological presentations. These external signs, such as xerosis, brittle nails, and alopecia, are not merely superficial but indicative of profound internal dysfunctions, including metabolic imbalances, immune suppression, and hormonal dysregulation—hallmark features of EDs. However, significant gaps in the literature remain. While many studies detail these dermatological manifestations, few explore gender-specific differences or variations among ED subtypes, such as binge–purge versus restrictive eating behaviors [69]. Research on the temporal resolution of these skin changes following nutritional rehabilitation is similarly sparse. Furthermore, despite robust evidence supporting their diagnostic utility, dermatological markers have yet to be fully integrated into routine clinical practice, particularly in psychiatry and dermatology, where interdisciplinary collaboration could significantly enhance patient outcomes [5,6].

The findings also highlight the bidirectional relationship between psychological and physical health in EDs. Dermatological changes, such as acne or hair thinning, can exacerbate body image concerns, intensifying disordered eating behaviors. Conversely, addressing these visible symptoms through targeted dermatological and nutritional interventions can have a positive psychological impact, improving self-esteem and encouraging recovery. This interplay indicates the importance of treating EDs holistically by considering both the physical and psychological dimensions of the disorder [43,66].

Beyond dermatological manifestations, the review draws attention to the critical relationship between nutrient deficiencies and psychiatric health. Essential nutrients like zinc, iron, selenium, and vitamins B, C, and D play pivotal roles in brain function and mental well-being. Deficiencies in these nutrients are linked to a range of psychiatric symptoms, including mood disorders, cognitive impairments, and exacerbated anxiety or depression, which are frequently observed in individuals with EDs. For instance, zinc is vital for neurotransmitter regulation and synaptic plasticity, while deficiencies in vitamins B6 and B12 can impair serotonin and dopamine synthesis, contributing to mood instability [3,4].

Clinically, this review advocates for a multidisciplinary approach to diagnosing and managing EDs, integrating dermatological, psychiatric, and nutritional perspectives. Dermatologists, in particular, are well positioned to identify subtle markers of EDs, such as Russell’s sign or lanugo-like hair, which may be overlooked by other specialists [41,54].

Close collaboration with nutritionists and mental health professionals can facilitate comprehensive care, addressing both the physiological and psychological aspects of these disorders. These insights also pave the way for more personalized treatment strategies, such as tailored nutritional supplementation and ongoing monitoring, ultimately improving patient outcomes [7].

Last but not least, existing conceptual frameworks connecting malnutrition, eating disorders, and skin health emphasize the critical role of nutrition in both prevention and treatment strategies. These frameworks have been increasingly enriched by the understanding of the skin–brain axis, which highlights the bidirectional communication between the skin and the CNS. This axis is influenced by neuroendocrine factors, immune signaling, and interactions with the microbiota, demonstrating the profound interplay between psychological stress, diet, and skin health. The skin–brain axis also sheds light on the mechanisms through which stress-related neuroendocrine activity exacerbates skin conditions. For instance, stress triggers activation of the hypothalamic–pituitary–adrenal (HPA) axis, leading to elevated cortisol levels that compromise skin barrier function and heighten susceptibility to inflammatory skin conditions such as atopic dermatitis and vitiligo. Furthermore, these neuroendocrine changes influence gut microbiota composition, establishing a more comprehensive framework known as the gut–skin–brain axis. This expanded model highlights the role of gut dysbiosis in promoting systemic inflammation, which contributes to skin disorders like psoriasis, acne, and atopic dermatitis. For example, in psoriasis, imbalances in gut microbiota intensify systemic inflammation through immune pathways mediated by cytokines such as IL-17 and IL-23 [71,72].

Applying these frameworks in clinical practice involves customizing nutritional interventions to address the unique needs of individual patients. Proper nutrition becomes a foundational pillar in managing these interconnected conditions and improving patient outcomes. Evidence suggests that dietary approaches such as the Mediterranean diet and gluten-free diets (GFDs) can be effective in psoriasis by reducing systemic inflammation, modulating gut microbiota, and mitigating immune dysregulation. Similarly, eliminating high-glycemic foods and dairy products has been associated with reduced acne severity, especially in cases where hormonal changes linked to the skin–brain axis are influenced by dietary patterns. Nutritional strategies to counteract inflammatory skin diseases resulting from HPA axis overactivation include diets enriched with antioxidants, omega-3 fatty acids, and vitamins D and E, all of which enhance skin health, mental well-being, and immune resilience. Incorporating fiber-rich diets promotes gut microbiota diversity and increases short-chain fatty acid (SCFA) production, which plays a crucial role in immune regulation and maintaining skin integrity. Probiotic supplementation has also proven beneficial for restoring microbiome balance and managing inflammatory skin conditions like acne and atopic dermatitis. Additionally, protein-rich diets support wound healing and collagen synthesis, making them an essential component of nutritional interventions for skin health [71,72].

By bridging gaps between dermatology, psychiatry, and nutrition, this integrated approach can better address the multifaceted challenges posed by EDs, providing a framework for effective, holistic care.

## 13. Conclusions

In conclusion, the nutrient–skin connection offers a valuable framework for understanding the systemic impact of EDs and improving their diagnosis and management. Towards this goal, dietary strategies like incorporating anti-inflammatory foods, avoiding dietary triggers (e.g., alcohol, trans fats, and high-glycemic foods), and supplementing with key nutrients (e.g., vitamins A, C, D, and omega-3 fatty acids) illustrate how nutritional science can bridge the gut–skin–brain connection. Probiotic-rich diets not only restore gut microbiota balance but also support mood regulation, reinforcing the synergistic relationship between skin health and psychological well-being. Also, by leveraging dermatological markers as diagnostic tools and fostering interdisciplinary care, clinicians can better address the complex interplay of physical and mental health challenges in patients with EDs. Future research should aim to expand our understanding of this relationship, ultimately paving the way for more effective, holistic treatment strategies.

## Figures and Tables

**Figure 1 nutrients-16-04354-f001:**
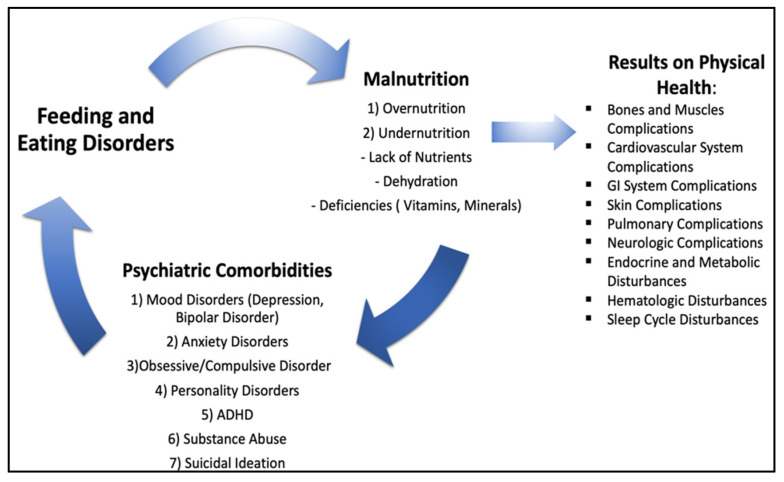
The vicious cycle of feeding and eating disorders, malnutrition, and physical health deterioration.

**Figure 2 nutrients-16-04354-f002:**
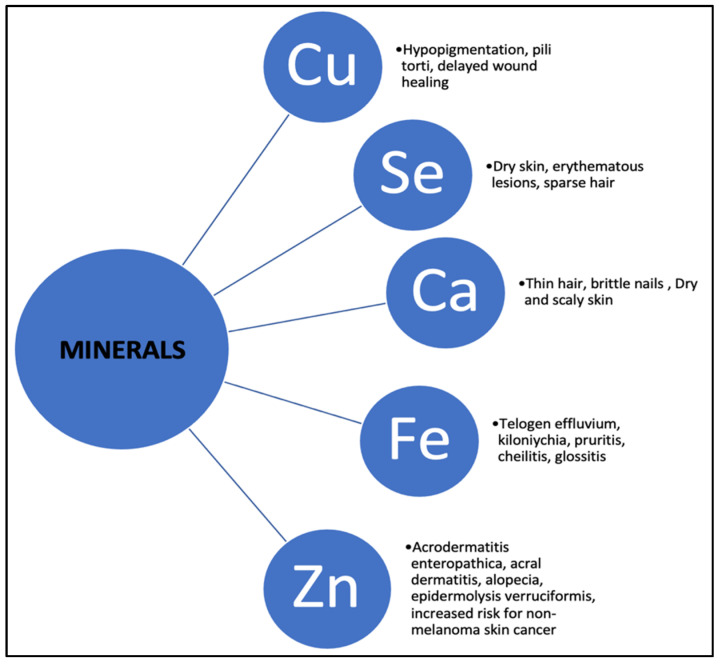
Skin signs of mineral deficiencies.

**Figure 3 nutrients-16-04354-f003:**
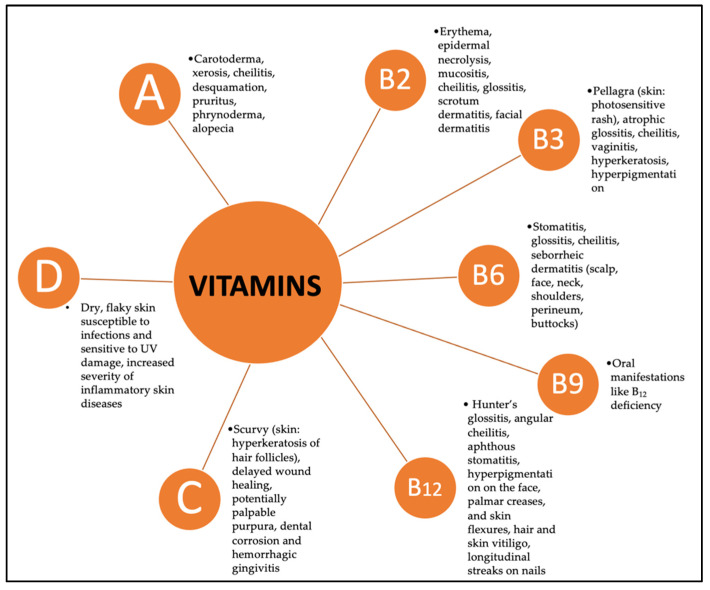
Skin signs of vitamin deficiencies.

**Figure 4 nutrients-16-04354-f004:**
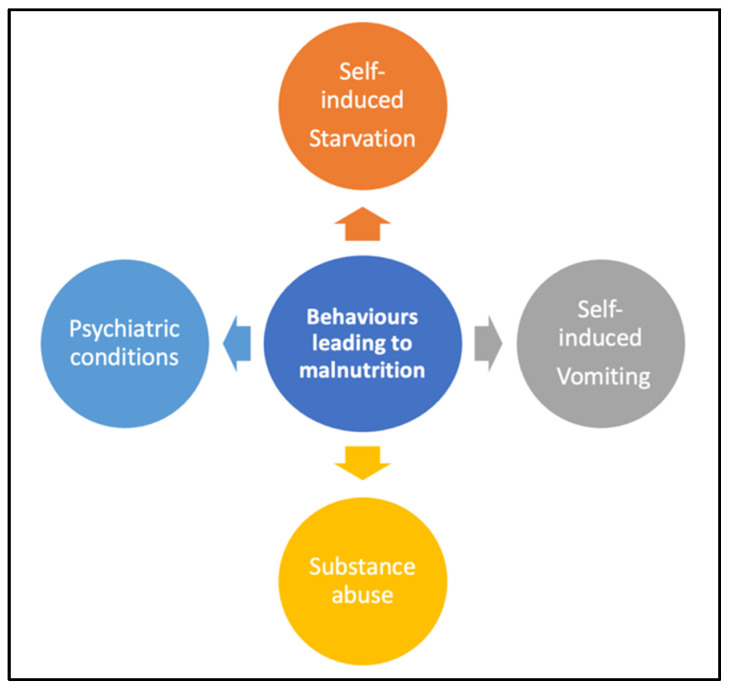
Behavioral patterns contributing to malnutrition.

**Table 1 nutrients-16-04354-t001:** DSM-5-TR diagnostic criteria for eating disorders [1].

Disorder	Diagnostic Criteria
Anorexia nervosa (AN)	- A. Persistent restriction of energy intake, leading to significantly low body weight. - B. Intense fear of gaining weight or persistent actions preventing weight gain despite being underweight. - C. Distorted self-perception of body weight/shape or lack of recognition of the severity of low weight. Specify Subtype: Restricting type: no binge eating/purging in the last 3 months; weight loss achieved through restrictive eating, fasting, or excessive activity. Binge-eating/purging type: recurrent binge eating or purging (e.g., vomiting and laxative misuse) in the last 3 months.
Bulimia nervosa (BN)	- A. Recurrent binge-eating episodes, defined by the following: Eating an excessive amount in a limited time. Feeling a lack of control during the episode. - B. Regular compensatory behaviors (e.g., vomiting and laxative misuse). - C. Episodes occur at least once a week for 3 months. - D. Self-worth is overly influenced by body shape and weight. - E. Behavior does not occur exclusively during anorexia nervosa episodes.
Binge-eating disorder (BED)	- A. Recurrent binge-eating episodes, as in BN. - B. Episodes involve three or more of the following: Eating rapidly. Eating until uncomfortably full. Eating large amounts when not physically hungry. Eating alone due to embarrassment. Feeling disgusted, depressed, or guilty afterward. - C. Significant distress about binge eating. - D. Episodes are not followed by compensatory behaviors and do not occur exclusively during anorexia or bulimia.
Pica	- A. Persistent consumption of non-food, non-nutritive substances for at least one month. - B. Behavior is developmentally inappropriate. - C. Not part of culturally or socially accepted practices. - D. If linked to another mental or medical condition, behavior warrants additional clinical attention.
Rumination disorder (RD)	- A. Repeated regurgitation of food over at least one month (food may be re-chewed, re-swallowed, or spit out). - B. Behavior is not due to medical or gastrointestinal conditions. - C. Does not occur solely during episodes of other eating disorders (e.g., AN, BN, BED, and ARFID). - D. If co-occurring with another mental disorder, symptoms require clinical attention.
Avoidant/restrictive food intake disorder (ARFID)	- A. Eating disturbance (e.g., lack of interest, sensory aversion, and fear of negative consequences) leading to the following: Significant weight loss or failure to meet growth expectations. Nutritional deficiencies. Reliance on supplements or enteral feeding. Psychosocial functioning impairment. - B. Not explained by food scarcity or cultural norms. - C. Not due to distorted body image and does not occur exclusively during AN or BN. - D. Not better explained by another condition or disorder.
Other specified feeding or eating disorder (OSFED)	- Applies when symptoms cause significant distress or impairment but do not meet full criteria for a specific eating disorder. Examples include the following: Atypical AN: all criteria for AN are met, except the individual’s weight is not below normal. Low frequency/limited duration BN or BED: criteria are met except for duration/frequency. Purging disorder: regular purging without binge eating. Night eating syndrome: recurrent night eating, with awareness, causing distress/impairment, not better explained by another condition.
Unspecified feeding or eating disorder (UFED)	- Includes cases with significant distress or impairment due to eating disorder symptoms that do not fully meet criteria for specific disorders. - Used when there is insufficient information or the clinician does not specify why criteria are unmet.

**Table 2 nutrients-16-04354-t002:** Medical complications of eating disorders.

System	Key Complications
Cardiovascular [15]	- Hypotension (<90/60 mmHg) and orthostatic hypotension - Bradycardia (<50 bpm day, <45 bpm night; hospitalization if <40 bpm) - QTc prolongation (>500 ms) with arrhythmia risk - Structural changes: reduced ventricular mass and output, valvular issues - Cardiomyopathy/heart failure risk from emetic abuse (e.g., ipecac syrup)
Gastrointestinal [15,16]	- Oral: tooth erosion, parotid enlargement, xerostomia, and gingivitis - Esophageal: reflux, ulcers, and rupture - Stomach/intestine: gastric dilation, delayed emptying, and colon issues
Hematologic [17,18]	- Decreased plasma volume, leukopenia, mild anemia, and thrombocytopenia - Bone marrow hypoplasia or gelatinous transformation
Bone health [19]	- Osteopenia to osteoporosis, linked to early-onset EDs and amenorrhea - Reduced lean body mass, reversible with weight restoration
Electrolyte imbalances [20]	- Common: hypokalemia, hyponatremia, and metabolic disturbances - Severe cases worsen ED diagnosis likelihood
Sleep disturbances [21]	- Insomnia and early awakening - Night eating and sleep-related eating disorders
Systemic impact [22]	- Multi-system effects: cardiovascular, GI, hematologic, neurological, and endocrine - Teeth, bones, and metabolism also impacted - Frequent lab abnormalities: anemia, electrolyte imbalances, and high LDL

**Table 3 nutrients-16-04354-t003:** Common nutritional deficiencies in eating disorders and their skin and mental health manifestations.

Deficiency	Underlying Cause of Deficiency in Eds	Results of the Deficiency on Skin and/or Mental Health
Minerals		
Copper (Cu) [26,27]	Low intake, especially proteins Malabsorption	Decreased pigmentation of skin and hair Delayed wound healing Pili torti Severe neurological complications
Selenium (Se) [28,29,30,31,32]	Low intake Alcohol abuse Malabsorption	Cognitive decline Depression Skin discoloration Xerosis Erythematous scaly papules and plaques
Calcium (Ca) [30,33,34,35]	Low intake Malabsorption Parathyroid Disturbances	Thin hair and brittle nails Dry, scaly skin Irritability and anxiety
Iron (Fe) [3,4,27,30]	Low intake, especially in a vegan diet Hemolysis of RBCs due to heavy exercise Malabsorption	Telogen effluvium (may lead to alopecia) Koilonychia Pruritus Cheilitis Glossitis Cold intolerance Sideropenic anemia
Zinc (Zn) [3,4,27,30,36]	Low intake Excessive exercise Laxatives (diarrhea) Malabsorption	Impairment of absorption of vitamin A Acrodermatitis enteropathica Psoriasis-like dermatitis affecting extremities Paronychia Mucosal findings (glossitis, cheilitis, and stomatitis) Hypogeusia (reduced taste sensitivity) Exacerbation of depressive symptoms Cognitive decline.
Vitamins		
A (retinol) [4,27,30,32]	Low intake Zn imbalance Lack of fatty acids, which help its absorption High intake from excessive consumption of carotenoids	Ocular complications Phrynoderma (skin like a toad) Carotenoderma (yellow-orange pigmentation) Xerosis Cheilitis Skin peeling Pruritus Alopecia
B1 (thiamin) [3,30]	Low intake, especially carbohydrates Alcohol abuse	Korsakoff syndrome Exacerbation of psychiatric symptoms Neurological problems
B2 (riboflavin) [3,4,27,30]	Low intake Potentially as a result of impaired thyroid hormone metabolism	Intense red erythema Epidermal necrolysis Mucositis Cheilitis and angular stomatitis, often with papules at the corners of the mouth that may bleed Glossitis Dermatitis of the scrotum in men Facial dermatitis Depression
B3 (niacin) [4,27,30,37]	Low intake Alcohol abuse	Pellagra 4 Ps: dermatitis, diarrhea, dementia, and, in severe cases, death Atrophic glossitis, cheilitis, vaginitis Hyperkeratosis and hyperpigmentation Psychosis, mania, and delirium
B6 (pyridoxine) [4,27,30]	Low intake Alcohol abuse	Stomatitis, glossitis, and cheilitis Seborrheic dermatitis affecting the face, scalp, neck, shoulders, and perineum Pellagra-like dermatitis (severe cases) Impairment of serotonin and appetite regulation Alteration in mental status
B12 (cobalamin) [4,27,30]	Low intake	Megaloblastic anemia Hunter’s glossitis Angular cheilitis Aphthous stomatitis Hyperpigmentation on the face, palmar creases, and skin flexures Hair and skin vitiligo Longitudinal nail streaks Neuropsychiatric symptoms Cognitive changes
C (ascorbic acid) [3,4,27,30]	Low intake	4 Hs: Hemorrhagic signs, hyperkeratosis of hair follicles, hypochondriasis, and hematologic anomalies Delayed wound healing Potentially palpable purpura Dental corrosion and hemorrhagic gingivitis
D (calciferol) [4,30,32,38]	Low intake	Dry, flaky skin susceptible to infections and sensitive to UV damage Increased severity of inflammatory skin diseases Alopecia (rare) Depression and/or anxiety Irritability and other mood disturbances Sleep disturbances Cognitive decline

**Table 4 nutrients-16-04354-t004:** Skin manifestations and associated eating disorders.

Manifestation	Description	Associated Eating Disorders	Resolution
Lanugo-like hair growth [41]	Fine, soft hair on the back, abdomen, face, neck, and arms. Linked to reduced 5-alpha-reductase activity and hypothyroidism.	AN	Gradual reduction with nutritional recovery [41]
Xerosis (dry skin) [43,44]	Dry skin caused by dehydration, reduced sebaceous activity, and lipid depletion from starvation or purging behaviors.	AN, BN	Requires nutritional recovery; emollients provide temporary relief.
Acrocyanosis [41]	Bluish discoloration of hands and feet due to vascular changes. Often accompanied by pallor and reduced pulse rate.	Severe AN	Reversible with weight restoration.
Pruritus (itching) [45]	Persistent itching, worsened at lower body weight; may stem from xerosis, malnutrition, or psychological factors.	AN	Resolves with weight and nutritional recovery.
Purpura [18]	Diffuse erythematous lesions due to thrombocytopenia and capillary fragility from malnutrition.	Severe AN	Resolves with weight restoration.
Russell’s Sign [54]	Calluses on knuckles from repeated self-induced vomiting.	Purging-type AN, BN	Lesions reduce with cessation of purging.
Carotenoderma [43]	Yellowish discoloration of palms, soles, and nasolabial folds due to excessive carotenoid intake.	Restrictive AN	Resolves with nutritional recovery.
Reactions to laxatives/diuretics [41,58]	Hives, lichenoid eruptions, and xerosis caused by misuse of laxatives or diuretics.	Purging-type AN, BN	Reversible with cessation of abuse.
Herpetic whitlow/Jehany Sign [55]	Painful pustules on fingertips or plantar hand surfaces due to repeated self-induced vomiting or HSV infection.	Purging-type AN, BN	Improves with cessation of vomiting.
Dental erosion [50,60]	Enamel erosion caused by frequent exposure to stomach acid from vomiting.	Purging-type AN, BN	Requires dental care and reduced purging.
Sialadenosis [56]	Noninflammatory enlargement of parotid glands due to chronic vomiting and altered salivary function.	BN	Resolves with cessation of vomiting.
Petechiae and subconjunctival hemorrhages [57]	Tiny red/purple spots or eye bleeding due to vomiting-induced pressure.	Purging-type AN, BN	Resolves with cessation of purging.
Self-harm scars [63]	Scars from burns, excoriations, or self-inflicted injuries, often linked to psychological distress.	AN, BN	Requires psychological and medical support.
TTM [62]	Compulsive hair-pulling causing prominent hair loss co-occurring with obsessive–compulsive tendencies.	AN, BN	Addressed through psychological intervention.
Hair loss and abnormalities [43,53]	Hair thinning, telogen effluvium, brittle or opaque hair due to protein and nutrient deficiencies.	AN (especially bulimic subtype)	Improves with recovery and adequate nutrition.
Nail abnormalities [43,50]	Includes fragility, periungual erythema, onychoschizia, koilonychia (spoon-shaped nails), Beau’s lines, and discoloration from B12 deficiency.	AN, BN	Improves with nutritional recovery.
Oral cavity issues [52]	Cheilitis, angular cheilitis, gingival bleeding, geographic tongue, and aphthous ulcers linked to malnutrition and vitamin deficiencies (e.g., riboflavin).	AN, BN	Requires nutritional recovery.
Erythema ab igne (EAI) [65]	Pigmented, net-like skin condition caused by prolonged exposure to low-level heat (43–47 °C) in individuals with anorexia nervosa. Often results from heightened cold sensitivity and reduced core body temperature.	AN	Resolves after discontinuing heat exposure.

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
