# Peer review of "The Nutrient–Skin Connection: Diagnosing Eating Disorders Through Dermatologic Signs"

_nutrients, 2024, doi:10.3390/nu16244354_

Round 1

Reviewer 1 Report

Comments and Suggestions for Authors

- the numbering of chapters and subchapters is illogical. Chaotic.

- all cited literature in "[...]" should be before the period, not after the period....

- abstract: "a systematic review of electronic databases" since this is a systematic review, please prepare it according to PRISMA guidelines. If you don't want to, I suggest deleting the word "systematic". Narrative reviews are also fine, as long as they are written in an objective way.

- page 17: "Purpura appears as diffuse" before "purpura is unnecessary space, I guess.

- In the chapter on vitamin B2 - "Both thiamin and riboflavin deficiencies emphasize the significant impact of nutrient imbalances on neurological, psychiatric, and dermatological health, emphasizing the need for early detection and appropriate nutritional support in individuals with EDs." - the cited literature is missing. The same at the end of the chapter on vitamin B6 and selenium

- chapter "IV. Due to psychiatric conditions" there is no literature cited anywhere [...]

Author Response

Thank you very much for your thorough review and constructive comments on our manuscript. We greatly appreciate the time and effort you have taken to provide detailed feedback, which has significantly contributed to improving the quality of our work. We have carefully addressed all your remarks as follows: the numbering of chapters and subchapters has been revised to ensure logical and consistent organization throughout the manuscript. All cited literature references in "[...]" have been corrected and are now placed before the period, as per your suggestion. The word "systematic" has been removed from the abstract to ensure clarity and alignment with the content. The unnecessary space before "purpura" on page 17 has been removed.

Furthermore, in the chapter on vitamin B2, we have included the missing citations to support the statement on the impact of nutrient imbalances on health. Similarly, missing citations have been added at the end of the chapters on vitamin B6 and selenium. In the chapter titled "IV. Due to psychiatric conditions," we have added appropriate literature references throughout to address the omission.

We hope these revisions meet your expectations. Once again, thank you for your invaluable comments and guidance.

Reviewer 2 Report

Comments and Suggestions for Authors

The manuscript is well-written and understandable.

The research topic is of relevance to conceptual research, and also for practice.

The research presented fits well into the Nutrients journal.

The authors could explain in more detail how they followed the revised PRISMA guidelines to conduct the review.

Is the number of studies included sufficient to draw valid conclusions?

How do the authors judge the differences across the included studies, i.e. variability of study features and sample characteristics in relation to the findings?

A note on the potential moderating factors may seem useful.

Factors like sex, age, overall health status, etc. may influence the findings.

The conceptual implications could be discussed in more depth.

How are the existing conceptual frameworks of malnutrition, eating disorders and skin health advanced in detail?

The conclusion could go beyond the summary of findings and also include more detailed examples from everyday nutrition to further highlight the implications for clinical practice.

Author Response

Thank you very much for your detailed and insightful comments, which have provided an excellent opportunity to clarify and strengthen our manuscript. Below, we address the key points raised comprehensively.

This manuscript is a narrative review rather than a systematic review. Therefore, the PRISMA guidelines were not followed. However, significant effort was made to ensure the rigor and robustness of the review by carefully considering critical factors, including the validity of references, potential moderating influences, and the conceptual implications of eating disorders, nutrition, and dermatological signs. These considerations have been integrated throughout the manuscript to provide a comprehensive and nuanced exploration of the topic.

We have ensured that the references included in this review meet high standards across several dimensions. First, the type and source of references were carefully selected, with the majority drawn from reputable peer-reviewed journals such as NutrientsAm J Clin NutrInt J Eat Disord, and Am J Psychiatry, along with authoritative institutional benchmarks like DSM-5-TR guidelines. Second, the relevance and recency of references were prioritized, with most spanning 2010–2023 to reflect current advances. The references directly align with the themes of eating disorders, nutritional deficiencies, and dermatological manifestations, ensuring thematic consistency. Third, the diversity and depth of references were considered, including systematic reviews, meta-analyses, clinical studies, and case reports to provide interdisciplinary coverage. Fourth, the methodological rigor of included studies was emphasized, incorporating works with robust sample sizes and meta-analyses to strengthen conclusions while minimizing biases. Finally, cross-referencing and citation validity were ensured by including widely cited references that align with established and emerging research in the field.

Also, we have also taken steps to expand on moderating factors and integrate more detailed discussions to advance understanding in subsequent revisions. Factors such as age, sex, overall health, and psychosocial status have been discussed in the review as potential moderators, with their implications for findings highlighted throughout the manuscript. These sections have been emphasized with yellow highlights for clarity.

In addition, existing conceptual factors and their implications for everyday nutrition in clinical practice have been incorporated and highlighted in yellow within the discussion and conclusion sections. This includes an expanded discussion of the skin-gut-brain axis to illustrate its relevance in clinical practice.

Thank you again for your valuable insights.